# Field-Emission Energy Distribution of Carbon Nanotube Film and Single Tube under High Current

**DOI:** 10.3390/nano14100888

**Published:** 2024-05-20

**Authors:** Lizhou Wang, Yiting Wu, Jun Jiang, Shuai Tang, Yanlin Ke, Yu Zhang, Shaozhi Deng

**Affiliations:** State Key Laboratory of Optoelectronic Materials and Technologies, Guangdong Province Key Laboratory of Display Material and Technology, School of Electronics and Information Technology, Sun Yat-sen University, Guangzhou 510275, China; wanglzh9@mail2.sysu.edu.cn (L.W.); wuyt79@mail2.sysu.edu.cn (Y.W.); jiangj97@mail2.sysu.edu.cn (J.J.); tangsh58@mail.sysu.edu.cn (S.T.); stskyl@mail.sysu.edu.cn (Y.K.)

**Keywords:** carbon nanotube, cold cathode, field emission, energy distribution

## Abstract

A narrow energy distribution is a prominent characteristic of field-emission cold cathodes. When applied in a vacuum electronic device, the cold cathode is fabricated over a large area and works under a high current and current density. It is interesting to see the energy distribution of the field emitter under such a working situation. In this work, the energy distribution spectra of a single carbon nanotube (CNT) and a CNT film were investigated across a range of currents, spanning from low to high. A consistent result indicated that, at low current emission, the CNT film (area: 0.585 mm^2^) exhibited a narrow electron energy distribution as small as 0.5 eV, similar to that of a single CNT, while the energy distribution broadened with increased current and voltage, accompanied by a peak position shift. The influencing factors related to the electric field, Joule heating, Coulomb interaction, and emission site over a large area were discussed to elucidate the underlying mechanism. The results provide guidance for the electron source application of nano-materials in cold cathode devices.

## 1. Introduction

Field emission is a type of electron emission that offers several advantages, including high current density, operation at room temperature, narrow energy distribution, and low power consumption. These characteristics make it suitable for use as the electron source in vacuum electronic devices (VEDs) [1]. Among the existing field-emission nano-materials, carbon nanotubes (CNTs) are among the best due to their extraordinary structure and physicochemical properties [2,3,4], such as a high aspect ratio, nano radius of curvature, high electronic/thermal conductivity, and robust chemical/thermal stability [5,6,7,8]. The CNT emitter has been shown to exhibit high current and high current density field-emission characteristics [9]. A single CNT can emit 10 μA of current with a current density of 2.04 × 10^4^ A/cm^2^ [10]. A CNT film can emit 0.34 mA of current, while the current density is much lower at approximately 137.91 A/cm^2^ [11]. The CNT emitter has also been proven to have a low energy distribution [12]. For example, Bonard. et al. [13] demonstrated that electrons emitted from carbon nanotubes feature an outstanding monochromatic energy distribution, as narrow as 0.18 eV.

In various applications, VEDs [14,15,16], including travelling wave tubes, X-ray tubes, electron microscopes, etc., all require a high current and high current density electron source to achieve optimal device performance. They also require a low energy distribution to improve electron beam control, such as focus, diffusion, laminar flow, etc. This raises the question of whether the CNT emitter can maintain its low energy distribution under high current and high current density, which has rarely been shown in the literature. In the literature, most of the energy distribution measurements have primarily been carried out under very low current conditions and emitted from a single CNT tip. Secondly, in devices, most CNT emitters are fabricated as thin films that consist of billions of CNTs. Thus, the second question is whether or not the CNT emitter can maintain its low energy distribution in a type of CNT thin film, which is important for device application. These two questions both have implications for the performance of the VED and should be investigated and clarified to provide guidance for device design and performance evaluation.

The aim of this study was to elucidate the variation in energy distribution spectra of a CNT emitter with changes in emission current and sample area. The energy distribution spectra of two types of emitter, including single and thin-film CNTs, were measured to evaluate the influence of the emission area. The focus was on the energy distribution of CNTs under high emission current working conditions. The underlying physics mechanisms were also discussed, thus offering insights into their potential for high-current and larger-area electron source device applications.

## 2. Experimental

### 2.1. Method of Measuring the Energy Distribution of Field-Emission Electrons

The retarding potential method [17] is adopted to measure energy distribution, which generates a decelerating electric field by applying a negative bias voltage, thereby creating a barrier that only allows electrons with energy exceeding this threshold to pass. The assembly, as shown in Figure 1, comprises an anode electrode that drives the electron emission of the cathode, a focusing electrode that focuses the electrons, and a Faraday cup that forms a deceleration electric field. Electrons emitted from the cathode go through the anode aperture and enter the deceleration field, undergoing a reduction in velocity. Electrons that successfully pass through this barrier are subsequently collected by the Faraday cup and transformed into a quantifiable current signal.

In the detailed procedure, a periodic triangular wave signal ranging from 0 to −20 V at a 1 Hz frequency (RIGOL DG4162, Agitek ATA-214) was applied to the cathode. The anode electrode was a metal plate covering a fluorescent film with a 1 mm-diameter hole at its center as the electron passage. The anode was held at a constant voltage during testing. Electrons passing through the hole were collected by the Faraday cup and then converted into a voltage signal using an oscilloscope (RIGOL DS2072A) and a picoammeter (KEITHLEY 6485). By simultaneously recording and analyzing the output voltage signal from the Faraday cup, which collects the current, and the input triangular periodic voltage signal, the corresponding electron energy distribution can be derived.

### 2.2. Preparation of CNT Emitters

Two types of CNT emitter were prepared: a vertically aligned CNT film and a single CNT. The vertically aligned CNT film was synthesized on a silicon substrate via thermal chemical vapor deposition (TCVD). The procedure commenced with the sputtering deposition of a 1.5 nm-thick iron catalyst layer on a silicon substrate. Subsequently, the substrate was placed in a quartz furnace and heated to 750 °C under an atmosphere of argon (400 sccm), hydrogen (100 sccm), and ethylene (50 sccm). After 60 min of growth, a dense array of vertical CNT cylindrical pillars was produced, achieving an ultralong height of 1.2 mm. Figure 2 depicts the morphology and structure of the synthesized vertically aligned CNTs. The diameter of a typical CNT is approximately 8–10 nm, and the single CNT examined in this study was a few-walled CNT with good crystallinity.

The single CNT was cut off from the vertical CNT film and transferred to the tip of a tungsten needle. First, the tungsten needle was moved to touch a CNT. Then, a high current was applied to pass through the CNT and needle. The large contact resistance between them generates Joule heat, allowing the CNT to weld firmly onto the needle. Finally, the CNT was detached from the film by applying a mechanical force, yielding a single CNT cathode. As illustrated in Figure 3a, the single CNT has a diameter of 25 nm and a length of 301 nm.

## 3. Results and Discussion

### 3.1. Field-Emission Energy Distribution of a Single CNT

Before conducting energy distribution measurements, the field-emission current–electric-field characteristics of the single CNT were first measured, as shown in Figure 3b. According to the Fowler–Nordheim equation [18]:(1)J=aE2ϕexpbv(y)ϕ32E,
where ϕ is the work function, *a* = 1.5414 × 10^−6^, *b* = −6.831 × 10^7^, *y* = 3.7947 × 10^−4^E^1/2^/ϕ, and v(y) can be approximated as $v(y)=\cos(\frac{1}{2}\pi y)$. Further modifications yield the FN characteristic curve shown in the inset of Figure 3b.

Testing was conducted in a 6 × 10^−6^ Pa vacuum, with the distance between the cathode and anode maintained at 7.3 mm. The single CNT demonstrated the capacity to emit currents up to 10.12 μA. The corresponding current density was 2.06 × 10^6^ A/cm^2^. This is the maximum current and current density that a CNT can sustain, which has been confirmed in our repeated experiment. If CNTs exceed this value, they will undergo vacuum breakdown and burn out due to the Joule heating effect. In the inset of Figure 3b, it can be observed that the FN curve is divided into three regions [19]. In Region I, under lower electric-field conditions, the field enhancement factor is primarily influenced by the geometric morphology of the CNTs, and the contribution of space charge effects to field emission is relatively small. Field-emission characteristics in this stage reflect the initial emission behavior of CNTs under low electric fields. In Region II, as the electric-field strength increases, the curvature of the FN curve decreases, mainly due to the emergence of space charge effects with increasing current density. In Region III, the slope of the FN curve increases again as the electric field further intensifies. The current density reaches a sufficiently high level that gases near the anode become ionized, producing significant ionic and electronic currents. Under the influence of the electric field, these currents separate and become part of the field-emission current, further enhancing the emission capability.

Thus, it is interesting to elucidate the field-emission electron energy distribution under such a high current situation. The energy distribution of the single CNT was measured across a range of current values, spanning from high to low, including 0.53 μA, 0.93 μA, 2.02 μA, 4.67 μA, and 10.12 μA, as shown in Figure 4a. It was observed that the peak of energy distribution shifts leftward and the full width at half maximum (FWHM) broadens as the current increases along with an anode voltage increase. Specifically, with a current of 0.53 μA at 500 V the peak was located at the energy band below Fermi level E-EF = −0.29 V, with an FWHM of 0.71 eV. Upon reaching a current of 10.12 μA at 700 V, the peak moved to E-EF = −0.93 V, and the FWHM expanded to 1.35 eV, which is 1.9 times larger than that observed with a current of 0.53 μA.

The peak shifting leftward indicates that most of the electrons emit from a lower energy state. In this experiment, the increasing emission current generates stronger Joule heat at the tip of the CNT and results in a temperature much higher than 2000 K [20,21,22]. In such a high temperature, more electrons have enough energy to transit to a high level; consequently, the Fermi level slightly down-shifts to a lower state. Because most of the free electrons emit from the Fermi level, the emission electron energy distribution peak should also down-shift with the Fermi level, which is reflected in the curve as a left-shift [19]. Another reason, as reported, is that the resistance of the CNT decreases with an increase in temperature, which is further facilitated by electric-field penetration and leads to a left-shift of the peak [19]. Another reported cause is a minor linear peak shift with extraction voltage, attributed to a Schottky barrier at the W tip contact [23].

The energy distribution peak broadens on both sides along with the increase in current, while left-side broadening is more prominent, as shown in the peak (color in mauve) at 10.12 μA (Figure 4a). The reasons for the peak broadening include three factors. Firstly, as illustrated in Figure 4b, referring to the left side of broadening, the increased electric field narrows the barrier width and reduces the potential barrier height, which increases the tunnelling probability of electron emission from lower energy states. The CNT examined in this study is a multi-walled CNT that can be regarded as metal and has a continual density of states near the Fermi level. The emission electrons from more energy states cause broadening of the electron energy spectrum at lower energy. Secondly [19], referring to the right side of broadening, the high temperature at the tip under high current flows increases the electron energy, and the electrons transit to a high level above the Fermi level in which the barrier width is narrower and the electron emission probability is even higher. Thus, this part of the emission electrons expands the width of the electron energy spectrum on the right side. Thirdly [24,25], the effect under high current density (in this experiment, 2.03 × 10^6^ A/cm^2^) that should be taken into consideration is the coulomb repulsion. The huge amount of emission electrons from the tip of the CNT into vacuum forms a high-density electron gas in front of the emitter, where the coulomb repulsion among electrons affects their energy distribution. The interactions among electrons scatter them and increase the transverse energy of the electrons, thus increasing the broadening of the electron energy distribution on both sides of the peak.

As shown in Figure 4c,d, the FWHM of the peak increases with current and electric field. Considering the reports by FRANSEN [26], the current emitted from a single CNT exhibits a width of approximately 0.3 eV in the electron energy spectrum at tens of nA. Due to the limitations of this experiment, the minimum emission current measured was 0.53 µA. Therefore, from the outset of the data collection, the Joule heating effect was incorporated, which broadened the distribution of electron energy. However, with a further increase in the current and electric field, the FWHM abruptly increases at approximately 700 V and 10.12 μA. As noted previously, 10.12 μA is the maximum current that a CNT can sustain. In such a situation, Coulomb collisions among electrons at high current densities further affect the broadening of the electron energy distribution.

In the literature comparison, as shown in Table 1, most of the works put concern on the small current region of field emission, which results in a narrow energy distribution. Even though they also reported the broadening of energy distribution phenomenon under relatively large currents, the data and explanations were not sufficient. In this work, the energy distribution during the full current cycle from the detected minimum current to the maximum current before vacuum breakdown were all measured, which provides a more comprehensive analysis for the subsequent CNT film measurements and furthers the electron source applications.

### 3.2. Field-Emission Energy Distribution of a CNT Film

The energy distribution of CNTs has primarily been investigated in single nanotubes. However, in devices, more cathodes are typically fabricated in the form of films. Thus, it is necessary to measure the energy distribution in a CNT film, which has been rarely reported. The methodological approach adopted in CNT films mirrors the one applied to the single CNT, ensuring consistency and comparability in the experiment.

The experiment employed a vertically aligned CNT film, cylindrical in shape, with dimensions of 1 mm in diameter and 1.2 mm in height. The CNT film emitter was grown on silicon wafers and mechanically transferred onto a stainless-steel substrate, as shown in Figure 5a. After laser etching around the edges of the cylindrical top, the effective emission area was approximately 0.585 mm², as illustrated in the inset of Figure 5a. The inset figure shows the morphology of the CNT film test structure, featuring a halfsphere emitter with a uniform electric-field distribution on the surface. Its field-emission current–field characteristics were first measured, as shown in Figure 5b. The measured maximum current was 2.06 mA at 5.0 V/μm, corresponding to a current density of 352.14 mA/cm^2^. It is noted that the current density is much smaller than that of a single CNT because the effective emission area is exponentially smaller in the thin-film sample. The fitted FN curve (inset of Figure 5b) showed a downward bending line, attributed to the space charge effect [18].

The energy distribution of the CNT film was measured at several current levels, ranging from 10.18 μA to 109.1 μA, and the voltage ranged from 600 to 1000 V, covering the low current range of the IE curve. Along with the increase in the current and voltage, the FWHM of the electron energy peak expanded from 0.51 eV to 1.70 eV, and the peak also shifted leftward, as shown in Figure 6a. Based on the above discussion on a single CNT, it is easy to understand the broadening of the energy distribution peak under high current. The peak at a low current of 10.18 μA showed a small FWHM of 0.51 eV and a symmetrical energy peak on both sides, which means that the film-type CNT emitter can maintain an excellent small energy distribution at low current. However, the FWHM of the peak parabolic increases with the current and voltage, as shown in Figure 6b. The FWHM at 109 μA increases to 1.7 eV, which is worse than that in the single CNT.

Firstly, the peak expansion on the left side is more obvious than that on the right side. Referring to the abovementioned explanation, it can be elucidated that the strong field-induced barrier width narrowing effect is more prominent than the high-current-induced Joule heating effect. The broadening of FWHM becomes saturated at a high current of approximately 109.1 μA, which is different to the single CNT, which demonstrates an abrupt increase at ever higher current. This is due to this current being far from the CNT thin-film’s maximum value. As the IE curve shows, the maximum current of the CNT film is 2.06 mA; thus, a current of 109.1 μA is in the low current range for the CNT film, which cannot reach the limit of the abrupt broadening of the energy peak. Secondly, besides the three effects discussed above, the additional factor is the large emission area in which the collected electrons are emitted from many nanotubes among the film. The diversity of the electron properties, including the emission angle and initial velocity, broadens the measured energy distribution. Although the field-emission electrons should have almost the same initial velocity according to FN theory, in a real emitter the surface electric-field distribution, nanotube tip morphology, nanotube resist, emission position on the film, and focus of the electro-optical system all have an influence on its velocity and, thus, the energy distribution. Thus, peak broadening is more significant than that observed for the single CNT. Thirdly, the Coulomb repulsion effect could also feasibly contribute to the broadening. Therefore, in the film CNT emitter the large emission area becomes the prominent factor of peak broadening. However, how much each factor contributed to the broadening was not elucidated in this study. More theorical calculations and simulations should be carried out to clarify the main mechanism(s) driving peak broadening.

In summary, the comparison of energy distributions between a single CNT and a CNT film clearly demonstrates that, at low current levels, both can maintain a narrow energy distribution, highlighting the advantage of field emission. As the emission current increases, the energy distribution of emitted electrons from the film follows a trend similar to that of a single CNT, but the rate of increase in distribution differs. Fortunately, this broadening does not occur as rapidly at high currents, only becoming pronounced when approaching the maximum current. However, it is important to note that a larger emission area may lead to an even wider energy distribution.

In practical applications, different devices have varying requirements for the quality of the electron beam [31], necessitating a trade-off based on specific circumstances. This balance is critical to ensure the effective operation of the device for its intended application. The small energy distribution of CNT film can meet the requirements of electron sources operating at low current. At high current, both energy distributions of single tube and film broaden in a parabolic manner with current exceeding 1 eV. For example, transmission electron microscopy requires a low electron energy distribution [32], while high-power microwave devices require high current [33].

Although the electron energy distribution broadens when CNT thin films emit high currents, they still have advantages over conventional thermionic cathodes. For example, for a sample area of 0.585 mm², the energy distribution of emitted electrons from the CNT film is less than 1 eV when the emission current is below 20 μA. In comparison, the thermionic cathode’s typical energy distribution width is 1 to 2 eV for emitting the same current [34,35]. This is due to the thermionic cathode needing to be heated to a high temperature to allow electrons to gain enough energy to get across above the surface barrier for emission. Temperature significantly affects the electron energy distribution in this case [36].

## 4. Conclusions

A comprehensive analysis using the retarding potential method was conducted on the field-emission energy distributions of both a single CNT and a vertical CNT film across a range of emission currents, spanning from low to high. At low current, both types of CNT had a small energy distribution of approximately 0.5–0.7 eV. At high current, the energy distribution expanded to 1.35–1.7 eV. The analysis revealed that the narrowing of the barrier width induced by the strong field accounted for left-side broadening of the peak, while the Joule heating induced by high current contributed to right-side broadening. Additionally, Coulomb repulsion also contributed to broadening on both sides. For CNT thin film, the diversity of electrons emitted from different CNTs further expanded the energy distribution. These results provide guidance for device application: there should be a tradeoff between the current and the energy distribution to reach a balance in device performance. Despite the expanded energy distribution at high current, the value of the CNT field emitter is still significantly better than that of a thermionic emitter.

## Figures and Tables

**Figure 1 nanomaterials-14-00888-f001:**
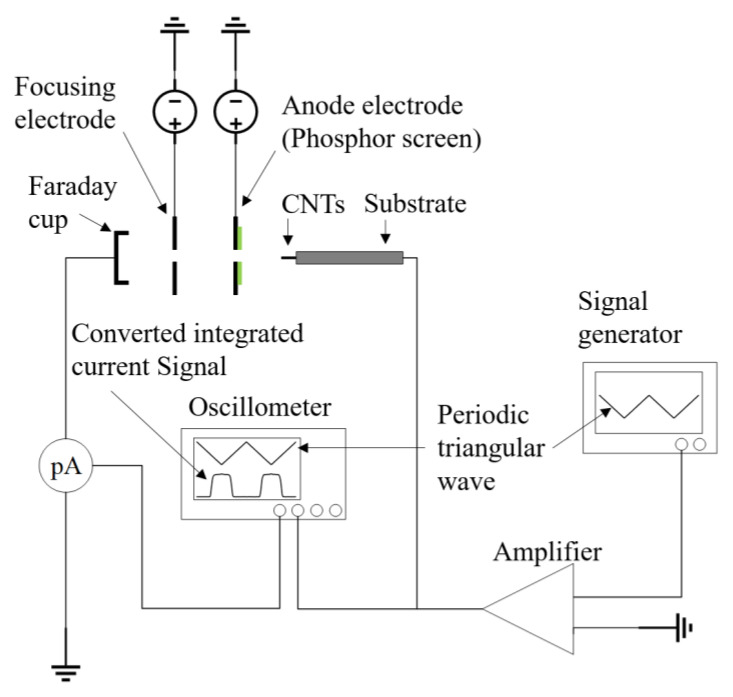
Schematic diagram of electron energy distribution measurement setup using the retarding electrode method.

**Figure 2 nanomaterials-14-00888-f002:**
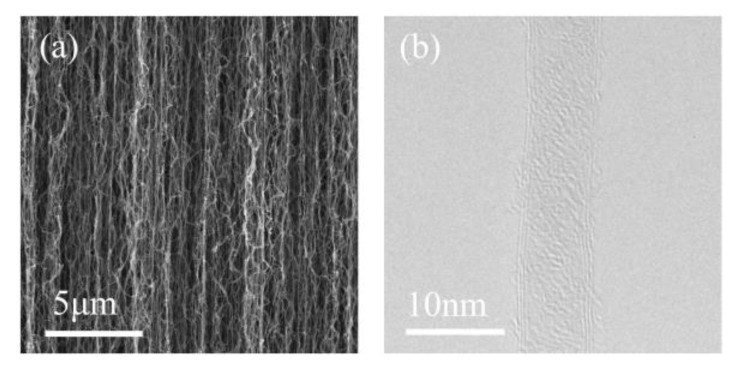
(**a**) SEM images of a vertically aligned CNT film; (**b**) TEM image of a single CNT.

**Figure 3 nanomaterials-14-00888-f003:**
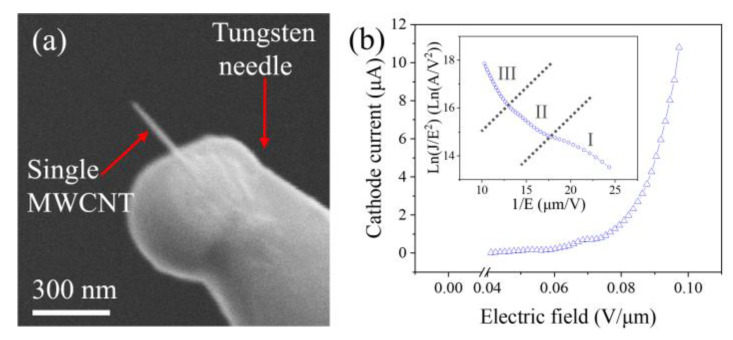
(**a**) SEM image of a single CNT mounted on a tungsten needle; (**b**) field-emission current and electric-field curve of a single CNT cathode, with an inset of the corresponding FN plot.

**Figure 4 nanomaterials-14-00888-f004:**
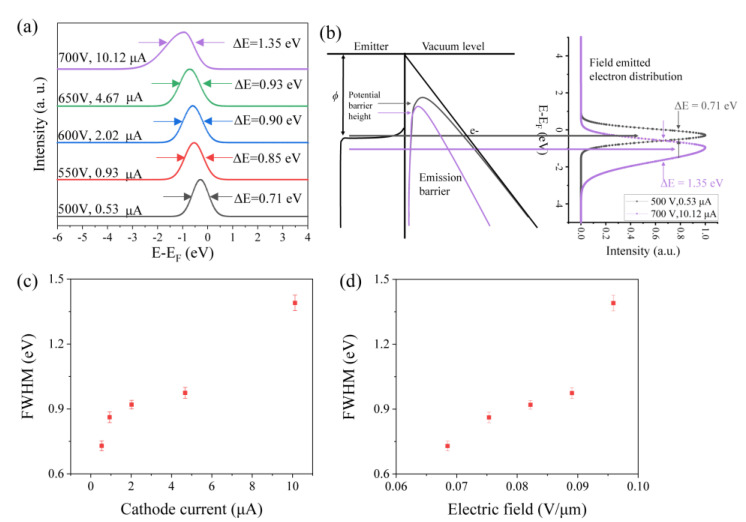
(**a**) Energy distribution spectrum of the single CNT varies with the signal voltage from 500 to 700 V and the emission current from 0.53 to 10.12 μA. The annotations indicate the anode voltage, cathode current, and FWHM of the energy distribution; (**b**) schematic energy band diagram of field emission, ΔE represents the FWHM; (**c**) FWHM relationship between energy distribution and cathode current; (**d**) FWHM relationship between energy distribution and electric field.

**Figure 5 nanomaterials-14-00888-f005:**
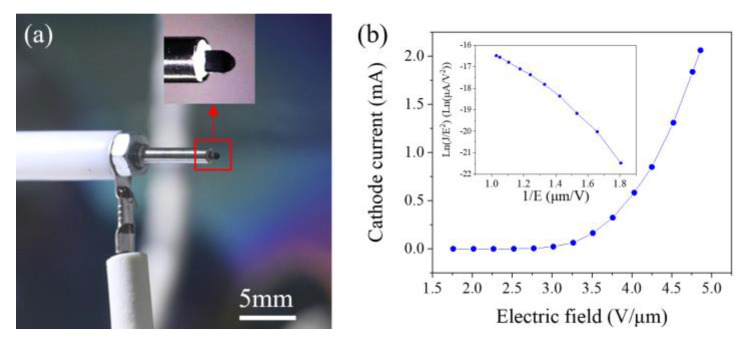
(**a**) Optical image of the physical structure in the test experiment, for which a CNT film cathode was mounted on a stainless-steel column, with an inset of the enlarged view of the CNT film test structure; (**b**) field-emission current and electric-field curve of the CNT film, with an inset of the corresponding FN plot.

**Figure 6 nanomaterials-14-00888-f006:**
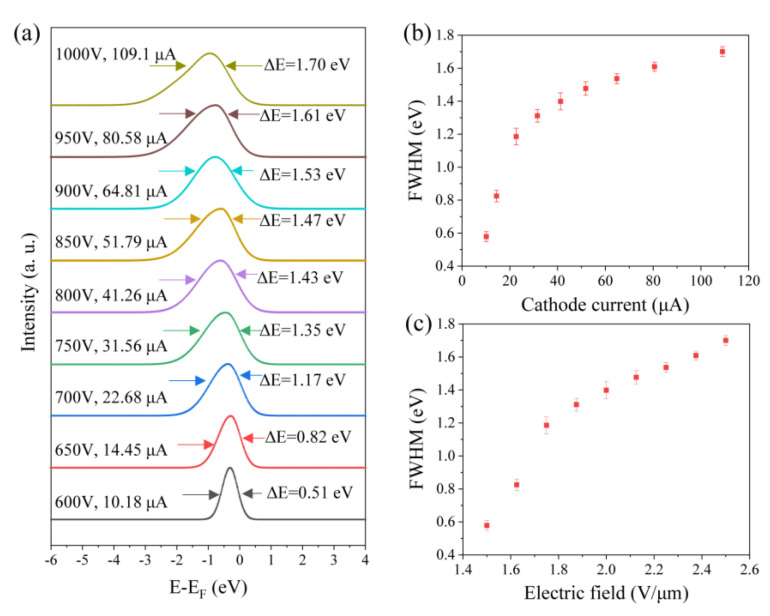
(**a**) Energy distribution spectrum of CNT film varies with an applied anode voltage of 600–1000 V and emission current of 10.18–109.1 μA, with annotations of anode voltage, cathode current, and FWHM of energy distribution; (**b**) FWHM relationship between energy distribution and cathode current; (**c**) FWHM relationship between energy distribution and electric field.

**Table 1 nanomaterials-14-00888-t001:** Literature comparison of field-emission energy distribution measured under high and low emission current regions.

	Low Current	High Current
Fransen M.J. [26]	0.11 eV @ -	0.75 eV @ 110 nA
De Jonge N. [27]	0.20 eV @ 10 nA	0.35 eV @ 500 nA
Hino S. [28]	0.3 eV @ 800 V	1.0 eV @ 1400 V
De Jonge N. [29]	0.20 eV @ 2.4 nA	0.30 eV @ 100 nA
Stephen T. Purcell [30]	0.2 eV @ 1 nA	0.9 eV @ 3.2 μA
This work	0.71 eV @ 0.53 μA	1.35 eV @ 10.12μA

## Data Availability

Data presented in this article are available upon request from the corresponding author.

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
