# Peer review of "Field-Emission Energy Distribution of Carbon Nanotube Film and Single Tube under High Current"

_nanomaterials, 2024, doi:10.3390/nano14100888_

Round 1
Reviewer 1 Report
Comments and Suggestions for Authors
This paper deals with an old subject, namely electron field emission by carbon nanotubes, in an attempt to help designers of vacuum electronic devices such as microwave amplifiers or portable X-ray tubes, by studying the trade-off to be made between the emitted intensity and the width of the energy distribution of the emitted electrons, in the case of an isolated MWCNT and a densely packed film of MWCNT.
My main concern with this paper is that the authors report some experimental results (that are sample dependent) but only interpret their results by some general considerations. They do not quote any generalized Fowler-Nordheim equation, as can be found easily e.g. in the Wikipedia page https://en.wikipedia.org/wiki/Field_electron_emission, nor use the classical Fowler-Nordheim equation in its validity interval to estimate e.g. the field enhancement factor and compare it for example with the simple approximation of the ratio of the length to the diameter, if the emitters are far enough from one another. Furthermore, since the experimental papers of De Heer et al., Rinzler et al. and Chernatozonski et al. in 1995, a lot of papers have addressed the problem of the calculation of the energy distribution of the emitted electrons by various kinds of nanotubes, including ab-initio calculations. Hence the results of this paper could probably be numerically compared to other experimental or theoretical results.
In other words: OK for the description of the results and how the authors got them, but what is the originality and significance of these results, with respect to the vast amount of literature already published on this subject since 1995? This point should be addressed both in their introduction and conclusion. Furthermore, the conclusion would probably also be enriched by what can be learned from the comparisons with other experimental or theoretical results requested above and by more quantitative rule of thumbs for the designers. The last but one sentence of the conclusion is not enough for me to warrant publication as is.
Finally, I note that there are many grammatical errors in the text, a few missing words and a few words that are not used with their proper meaning.
Comments on the Quality of English LanguageThere are many grammatical errors, a few missing words and a few words that are not used in their proper sense.
Author Response
please see the attached word file, thanks.

Reviewer 2 Report
Comments and Suggestions for Authors
The paper by Wang et al. deals with Field Emission Energy Distribution of Carbon Nanotube Film and Single Tube Under High Current. In this work, the energy distribution of a single carbon nanotube (CNT) and a CNT film was investigated in various current regions, along with discussions on the mechanism. The paper is well-written, with clear concepts and explanations. Therefore, I recommend accepting this manuscript. However, the authors need to address the following concerns:
(1) The statement “The small energy distribution of CNT film can satisfy the application of electron source which work in low current. While at high current level, both of the energy distributions broadening in a parabolic increase with current and large than 1 eV.” The grammar in this sentence needs some corrections for clarity.
(2) For the statement “the current and the energy distribution should be trade off to reach a balance of device performances”. It would be better to mention the tolerance levels of current and energy distribution for device applications, including references.
(3) The authors stated, "the value of the CNT field emitter is yet much better than that of a thermionic emitter." I recommend mentioning the drawback of the thermionic emitter in this context so that the reader can better understand the comparison.
Comments on the Quality of English Language
There is no issues about english.
